# A Genetic Programming-Based Low-Level Instructions Robot for Realtimebattle

**DOI:** 10.3390/e22121362

**Published:** 2020-11-30

**Authors:** Juan Romero, Antonino Santos, Adrian Carballal, Nereida Rodriguez-Fernandez, Iria Santos, Alvaro Torrente-Patiño, Juan Tuñas, Penousal Machado

**Affiliations:** 1CITIC-Research Center of Information and Communication Technologies, University of A Coruña, 15071 A Coruña, Spain; adrian.carballal@udc.es (A.C.); nereida.rodriguezf@udc.es (N.R.-F.); iria.santos@udc.es (I.S.); 2Department of Computer Science and Information Technologies, Faculty of Communication Science, University of A Coruña, Campus Elviña s/n, 15071 A Coruña, Spain; 3Department of Computer Science and Information Technologies, Faculty of Computer Science, University of A Coruña, Campus Elviña s/n, 15071 A Coruña, Spain; antonino.santos@udc.es (A.S.); alvaro.torrente@udc.es (A.T.-P.); jtunas@corunet.com (J.T.); 4Centre for Informatics and Systems of the University of Coimbra (CISUC), DEI, University of Coimbra, 3030-790 Coimbra, Portugal; machado@dei.uc.pt

**Keywords:** RealTimeBattle, genetic programming, robots, evolutionary robotics, evolutionary game, artificial intelligence, creative computation

## Abstract

RealTimeBattle is an environment in which robots controlled by programs fight each other. Programs control the simulated robots using low-level messages (e.g., turn radar, accelerate). Unlike other tools like Robocode, each of these robots can be developed using different programming languages. Our purpose is to generate, without human programming or other intervention, a robot that is highly competitive in RealTimeBattle. To that end, we implemented an Evolutionary Computation technique: Genetic Programming. The robot controllers created in the course of the experiments exhibit several different and effective combat strategies such as avoidance, sniping, encircling and shooting. To further improve their performance, we propose a function-set that includes short-term memory mechanisms, which allowed us to evolve a robot that is superior to all of the rivals used for its training. The robot was also tested in a bout with the winner of the previous “RealTimeBattle Championship”, which it won. Finally, our robot was tested in a multi-robot battle arena, with five simultaneous opponents, and obtained the best results among the contenders.

## 1. Introduction

In 2002, Fogel described a computer program that learns to play chess without any deep knowledge about the game, instead learning by playing against humans [1]. The program in question uses evolutionary computation techniques to evolve a chessboard evaluation strategy, which is then combined with standard artificial intelligence (AI) techniques to efficiently play chess. Similar approaches with somewhat different techniques and levels of complexity were applied to chess problems by AlphaZero and clones in the last ten years.

Following the same approach, in this work, our goal was to test the ability of AI systems (specifically, evolutionary computation) to obtain different effective strategies in the context of a tank combat simulator. The objective was to create programs to control the tank. The adversaries will be tank controller programs created by humans. No tactic, or any other information, was provided to the system. The only external information was that provided by the simulator in each battle.

Traditionally, developers have programmed robots using rules and control structures that include knowledge of the environment in which the robot will perform. Evolutionary robotics (ER) is a strategy for the design and control of robots in which the desired characteristics or behaviors are determined by a function that measures adaptability to the environment, without the need to program specific behaviors [2,3,4] (more information available at http://gral.ip.rm.cnr.it/evorobot/).

ER is related to the concept of evolutionary computing [5] (EC), developed according to the basic principles of Darwin’s evolution of species through natural selection (i.e., the most adapted living beings have the best chance to survive and reproduce, and therefore their characteristics are inherited by the next generation). EC comprises techniques such as evolutionary programming [5,6,7], evolutionary strategies [8], genetic algorithms [9] and genetic programming (GP) [10,11,12,13,14]. GP has been used extensively on conventional robots programming such us [15,16,17,18,19,20].

This article presents the development of an ER system in a RealTimeBattle (RTB) environment. RTB is an established multi-platform simulation environment that hosts competitions between virtual robots programmed with different techniques and languages [21] (more information available at http://realtimebattle.sourceforge.net). Each program controls a virtual robot that is trying to destroy each other. The programs receive input about the environment from radar and control firing of a cannon. The robots can move, accelerate, brake and turn, like ordinary vehicles on wheels. In this work, we describe a tool that, automatically and through GP, generates robots that could win in RTB competitions using low-level instructions (rotation and acceleration) instead of high-level instructions (such as attack or escape). The capability of the best-evolved robot in fighting multiple adversaries at the same time, was also studied.

This article is organized as follows: Section 2 presents the RTB domain in which the robots interact, some similar systems like Robocode, and related work; Section 3 defines the GP technique that was used to obtain our robots in this environment; and Section 4 describes the system that was proposed to this effect. The experiments and results are discussed in Section 5 and Section 6, and conclusions and directions for future development are presented in Section 7.

## 2. Realtimebattle Environment

In RTB, the final purpose of each robot is to destroy its enemies using only radar and a cannon. Each robot has a certain amount of energy. At the start of the match, all robots have the same amount of energy. When this energy reaches zero, the robot is eliminated.

The radar provides information on what is happening during the battle: each time the robot is updated, it receives a message informing about the nearest object in the current (linear) direction and about its distance and type of object. If a robot is detected, information is provided on its energy level.

The cannon is used to inflict damage on enemies. In RTB, a shot moves with a constant speed in the direction indicated by the cannon. Every shot moves until an impact with an object. When a projectile is fired, it acquires a certain amount of energy that determines the damage suffered by the target robot upon impact. However, firing itself entails a certain cost, since the robot loses an amount of energy proportionate to the energy of the shot; therefore, a robot cannot shoot repeatedly, because doing so would rapidly consume its energy. If the shot misses, it disappears upon collision with the arena border. We should also note that the cannon and the radar of each robot are separately mobile.

The battle arena, which is highly configurable, allows the players to define battlefields of different shapes and sizes. The arena, robots, and components are shown in Figure 1.

In RTB, the robots move like a car with wheels. Air resistance is applied in a direction opposing movement and increases with the robot’s speed. There are three ways a robot can change its movement: acceleration, rotation, and braking. Friction combined with acceleration makes the robot turn; when it brakes, friction reaches a maximal level that blocks the wheels and makes the robot slip instead of rotating, like a real car. Speed cannot be controlled directly—only by acceleration and braking. Acceleration is applied to increase speed in the actual direction of the wheels and is the only way the robot can move. Rotation makes the robot turn around, which affects only its frontal direction (the direction of the wheels), not its movement.

Two types of objects appear at random in the arena in the course of battle: mines and cookies. If a robot collides with a mine, it loses an amount of energy similar to receiving a shot, while a cookie provides energy when it is collected. Both types of object (cookies and mines) disappear after collision. A robot that collides with arena walls or rival robots loses less energy than when colliding with a mine. In a collision, robots rebound.

A wide range of RTB competitions, in which robots of all kinds face each other, are frequently organized. Spain, in particular, offers a series of annual competitions, such as *iParty*, *Euskal Encounter* and *Campus Party*, in addition to competitions organized through the official RTB website. The confrontations can be between two or more robots, but direct combat between two rivals remains the most popular formula.

There are other tank-fight simulation tools, including RobotBattle (more information available at http://robotbattles.com) and Robocode (more information available at http://robocode.sourceforge.net/). However, RobotBattle is only available for Windows and robots must be written in specific languages. Thus, the possibilities of developing intelligent robots are limited, since players are forced to learn the language, instead of using one they already mastered. More relevant, Robocode limits program size. Robocode has three categories, allowing robots of 250, 750 and 1500 bytes (in compiled form), so it is not possible to create very complex behaviours.

Robocode robots are written as event-driven Java or .NET programs and executed as threads in the main program. The main loop controls robot activities, which can be interrupted on different occasions, called events. In RobotBattle and Robocode, the simulation model and the actuators are the same: each tank can rotate its entire body, as well as its radar and cannon. The cannon and radar can move independently, as in RTB. The physics of both simulators both seek to be as realistic as possible in terms of rebounding, gravity, wind and resistance.

### Previous Works on RTB and Robocode Evolutionary Robots

Several researchers have attempted to automate the creation of simulated fighting robots for RTB and Robocode. For instance, Eisenstein [22] successfully applied genetic techniques to produce controllers based on Kaelbling’s REX language [23]. To determine the fitness of each evolved robot, a score was assigned based on how much damage it did to its opponent, plus a substantial bonus for being the last robot standing (in multi-tank matches). In the evolution stages, Eisenstein used 10 robots coded by humans for training. A robot called “SquigBot”, tagged as *expertize* for being in the top 4 in the Robocode League, was used for external testing. After 13 generations, a 100% success rate was achieved against the training robots, and a 50% success rate when fighting multiple adversaries was achieved after 60 generations. Eisenstein’s robots were able to defeat the external testing robot (a robot that they weren’t trained against) 50% of the time in one-on-one combat.

Gade et al. [24] used different techniques to develop different functions of their robot tank. They used an artificial neural network for shooting, using the output of this module to determine the angle to which the gun is turned and the probability of a successful hit. Reinforcement learning [25] was used for target selection, and GP for the movement and radar control. They used a small, self-designed language based on LISP to fit the application domain [26]. The fitness of each robot was determined by the number of points scored in 10 rounds, plus a bonus if the robot was the last survivor. With this hybrid architecture they developed a robot, Aalbot, which they tested in multi-tank fights against three other robots, achieving 2nd place on average. The opponents were a sample robot and two expertise robots, called “SquigBot” (also employed on [22]) and “Peryton”.

Hong and Cho [27] adopted genetic algorithms to produce different behavior styles for Robocode robots. They only used basic actions related to movement, gun and radar, environment info and events in the battle. They designed six primitive behaviors, classified as move-strategy, avoid-strategy, shoot-strategy, bullet-power-strategy, radar-search-strategy and target-select-strategy. They test their method in matches against three tanks provided by Robocode and against “BigBear”, champion of the 2002 Robocode Rumble [28]. Their method only achieved good results with the first three tanks, finding that different behaviors were needed to defeat each one, while the method could not find any suitable strategy to defeat BigBear.

Shichel et al. [29] described their first attempt to introduce robots created using evolutionary design into the Robocode League (later expanded upon by [30]). The strategy was predefined by the authors: to move the cannon (and radar) to the right until a tank was found, then move to a value (x), turn right one azimuth (y), and turn the barrel another azimuth (z). They used a GP system that only generated three values: x, y, z. So, they developed numerical expressions that were given as arguments to the players’ actuators. Two previous adjustments were made: the radar rotation command was omitted and the energy fire actuator implemented as a numerical constant. The authors used the score obtained by each player and the score obtained by its adversary to determine fitness. GPBot, the robot created using this method, participated at HaikuBot League in 2004. Among a total of 27 contenders, GPBot, the only robot not written entirely by a human, won third place. The authors did note some generalization problems when playing against other opponents, even relatively inferior ones.

Nidorf and Barone [31] compared two machine learning approaches—NEAT and accuracy-based learning classifier system (XCS)—to Robocode in various tactical challenges related to scanning and targeting Robocode tanks.

Harper [32] used grammatical evolution [33] together with spatial co-evolution in age-layered planes or SCALP [34]. They developed a Backus–Naur form grammar composed of 15 commands and eight functions related to movement, cannon and radar turn, position and miscellanies. Their experiments were run on a 15 × 16 SCALP grid (240 nodes) consisting of four layers. The system was allowed to evolve for 440 generations using 16 human-coded robots as reference. The resulting robot was able to defeat “Peryton” regularly and won 50% of battles against “SquigBot”.

More recently [35], Harper compared a spatial coevolution system in which robots train with each other with a system that used a handmade fitness gradient consisting of pre-selected human-coded robots. In both cases, the robots fell into the Robocode nano-bot category, which limits compiled code to 250 bytes. This requirement seriously limits the ability to use various GP techniques, but also limits human-coded robots: “the human-coded robots one might find in the leaderboards are a little less more sophisticated than in the larger categories”. Ref. [35] A system of grammatical evolution similar to that described in [32] was used in both cases, with similar results. The authors emphasize that the coevolutionary system demonstrates the same capabilities but did not require any human input. The behavior of the winning robots is relatively simple, according to the authors, mainly following the walls and firing “roughly”.

This paper describes the development of a robot for RTB using an evolutionary model based on GP. The approach allows each tank to be developed using different programming languages and does not limit the complexity of players. The resulting robot was tested in one-on-one fights and in free-for-all fights and showed some complex behaviors.

## 3. Genetic Programming

GP is based on Darwin’s natural selection principle, and although it is an extension of genetic algorithms [9], it represents the individuals of a population through programs (represented as trees) instead of strings with a fixed length. Any type of computer program can be seen as the successive application of functions to parameters.

The compilers use this characteristic to internally translate a given program into a syntactic tree, and then convert that tree into an executable machine code. In GP, it is vital to adequately define the set of instructions that can be used to solve the problem.

A GP algorithm (Figure 2) starts with an initial population of arbitrarily generated programs (“individuals”). These programs, or trees, consist of functions and terminals that are suitable for a specific problem and decided upon in each experiment by the designer of the system. Next, each individual of the population is classified by a *fitness* function that is defined by the programmer and obtains the aptitude of the individual in the course of its adaptation. As such, a new population is created by applying the genetic operators of reproduction, crossover and mutation (in a lesser degree) to individuals that are selected based on their aptitude, and the previous generation is replaced. As time goes by, the most adequate individuals survive, while those that provide the worst solutions to the problem disappear. This cycle continues for a set number of generations or until a program that solves the problem is found.

GP is a model induction method in that both the structure and parameters of the solution are explored simultaneously. The main advantage of GP is the ability to create comprehensive operator-based behaviors that can be tailored to a particular problem. In fact, the result of a GP system is a program that can be analyzed (although it can be arduous at times). In addition, the GP adapts well to complex domains in which little information is available, such as the one proposed in this work, in which the tank has little information about the environment. A GP system can even lead to behaviors beyond the programming capabilities of its creators and generate creative results. There are several patents created by GP [36].

## 4. Genetic Robots Programmer

The tool developed to create RTB robots, “Genetic Programmer”, has three subsystems. The first subsystem, in which the GP algorithm is implemented, launches the RTB battles and detects the fittest robot. Since each battle takes place in real time and lasts no more than two minutes, regardless of the computer’s power, the time used for the evaluation of one single generation of hundreds of individuals is significant. Even so, each battle requires only low CPU use, and several battles can take place at the same time. The evaluation time was reduced by employing a parallelization algorithm based on threads, so different individuals could be evaluated simultaneously.

Each thread was responsible for obtaining the adjustment of a robot by confronting it with different adversaries. The second subsystem provides the execution of the robots and all necessary support for the execution of the programs, which allowed users to evaluate the population’s individuals. RTB robots are executable files, launched by the environment as child processes at the beginning of a battle. From this moment on, the RTB controls the robot and sends messages to it concerning the state of the battle. The robots act based on the information contained in their programs. The last subsystem is the user’s communication interface, which allows defining the parameters that control the evolutionary process and to manage the process execution. Figure 3 offers a diagram of the system and its communications.

## 5. Evolution Toward a Competitive Robot

In this section, we are going to explore different experiments that try to get increasingly competitive simulated tanks. The experiments differ in the set of instructions used, in the rivals with which the robot is faced, in the calculation of fitness and in some essential parameters of the genetic programming engine. The final solution even uses two different fitness types (one simpler initially and one more complex in recent generations).

In each of our battles, one of the robots is an individual of the population that is being evaluated, and the other is one of the set of human-coded rivals (see below). All battles took place between two robots; even though RTB allows simultaneous battles between different robots, we chose a scheme of elimination battles with two opponents that is commonly used in competitions.

The adaptation of each robot was measured by confronting it with several human-coded rivals in a pre-established number of battles. The fitness function is the square of the sum of the points (score) obtained in each battle. The points depend on whether the robot has won (3 points), tied (1 point) or lost (0 points). To obtain the maximum score, the evolutionary robot must win all the battles against human-code rivals. The rivals and number of battles changed in each experiment. The rivals were built with conventional programming techniques and participated and won in several RTB competitions.

The rival robots and their functionalities are as follows:*pikachu*: Like most RTB robots, this one reduces its acceleration when close to the wall. The robot randomly sets a new rotation and speed periodically, which makes it difficult for the opponent to predict its path. When it sees its rival, it stops rotating and shoots. It eats cookies and destroys mines.*chimpokomon*: Initially, this robot spins without moving. If it sees a mine it shoots, if it sees a cookie, it moves toward it. If it sees a rival robot, it retreats and starts to shoot with a force that is proportional to the distance from the rival (the nearer the rival, the higher the energy). At the same time, it places its radar and cannon in sweeping mode, trying not to lose sight of the rival. When this robot receives an impact, it rotates at a certain angle to the direction of the impact. This robot placed second in the competition that took place in 2002 at the Faculty of Computer Sciences, University of A Coruña. It also participated in competitions organized by Xunta de Galicia (the local government) in 2001 (first place) and 2002 (second place).*falky*: When this robot finds its rival, it moves towards it and fires six shots. If it finds a cookie it tries to eat it. It only fires at mines that are very close or that move quickly enough for it to collide with. The most interesting aspect of this robot is its exploring character: if it spends a certain amount of time seeing nothing but the walls of the arena, it starts to move and explore its environment. As it approaches the walls, its acceleration decreases and rotation speed increases to avoid impact. If this robot runs into a rival or is hit, it turns toward the impact. Falky won the 2001 RTB competition at the Faculty of Computer Sciences in A Coruña.*rogynt*: As long as it observes the wall, this robot moves with a constant acceleration and rotation angle. When it finds a rival, it aligns its radar and cannon with the front of the robot and moves toward it, firing a set of 15 shots. If it has previously seen its rival but no longer does, it sets its radar and cannon in sweeping mode to find it. When this robot approaches the wall, it stops and spins 180 degrees. This robot won the 2002 RTB competition at the Faculty of Computer Sciences at A Coruña.

Four experiments are presented in this section. Each employed a different set of operators and terminal nodes that were used as basic pieces in the construction of the genetic program—an “instruction set”. The instruction set determines the possibilities of the robots built with it. For example, if there is no instruction to “move the radar”, this radar will always be fixed. On the other hand, a very large set of instructions will generate a very large search space. Considering that the experiments performed involved over 2000 h of CPU use, it is really relevant to correctly define the instruction set.

The experiments adjusted several parameters to obtain better solutions. Those parameters can be divided into two groups: those that belong to the GP’s own algorithm (population size, selection method for the individuals, etc.), and those that are directly related to fitness, such as the number of confrontations with each rival, points for each victory, etc.

### 5.1. First Instruction Set

In the initial instruction set, some of the functions possessed two child nodes. As such, the if-then-else structure was implemented and the left or right branch executed according to the fulfillment of the condition indicated by the name of the function. For instance the instruction “IfRobot” has two branches: the left branch is executed if a robot is detected in the radar, and the right branch otherwise.

In RTB, each action that can be carried out by the robot is associated with a certain amount of energy or potential: for instance, the robot can move more or less quickly. This is represented by giving each action a child node (parameter of the function) that indicates the strength with which the action is carried out. This parameter is represented according to the maximal strength for a determined action, which is a constant that is defined in RTB and varies for each action.

In this instruction set, we defined four terminals:“1” maximal strength, “3/4” three-quarters of the maximal strength, “1/2” half of the maximal strength, “1/4” one-quarter of the maximal strength. Table 1 shows the defined sets of functions. Table 2 provides the RTB and GP parameters of this experiment.

To illustrate the differences between the four sets of instructions, a simple example robot is shown for each. This example robot rotates the radar and the cannon simultaneously and fires as soon as it detects a mine. Figure 4 shows the codification of this robot using the first instruction set. Figures 6, 10 and 12 show the codification of the same behavior in the other three instruction sets.

Table 2 provides the parameters of the first experiment using this instruction set. As shown in the last line, fitness was calculated by performing 20 battles: five battles against each of the four rivals (pikachu, chimpokomon, falky and rogynt). Each won battle added 3 points to the score, while each draw added 1. Thus, the maximum raw fitness was 3600 ($60^{2}$, corresponding to 20 won battles).

In this experiment (and the following one) we used “Greedy Over-Selection”, which divided the population in two groups based on fitness. The first group contained individuals with better fitness (32%) and was selected more commonly (80% of the time). The second group included other individuals, selected at low rates. Based on results, the Greedy Over-Selection scheme was abandoned in experiments 3 and 4 in favor of a conventional, fitness-proportional roulette selection method that yielded better results. The fitness grew with each generation, but the final average fitness in this first experiment was below 0.3 (Figure 5).

An RTB battle has a certain degree of randomness—for example, the initial position and orientation of the robots are random. Thus, a test was performed in which the most successful robots fought 30 battles against each rival (120 in total). We did not obtain good results. Table 3 shows the application of a test of 30 battles to the robots of generation 15, which faced each rival that appeared during the training, including the number of battles won (W), tied (T) and lost (L). A tie means that the two robots were still alive when the battle time ran out. For a robot that has been put to test to be considered a solution, it must win at least half of its battles against each rival.

### 5.2. Second Instruction Set

In the first instruction set, the number of elements of the action set was excessive and redundant, especially after equipping each element with four strength levels. The second instruction set is a simplification of the first. Each action is associated with only two strength levels: the maximal and minimal level, which reduces the cardinality of the set, but maintains functionality. Moreover, the terminal (1/2, 3/4⋯) is included in the function “shot”, creating the functions “shotmin” and “shotmax”. For example, Figure 6 shows the program of a robot that is built, as in Figure 4, to destroy mines. To do so, it rotates the radar and the cannon with maximum speed and fires with minimum energy.

As an example of the use of this new instruction set, we show the results of an experiment that trained the robots to fight only the rival that was considered strongest, *chimpokomon* (Table 4). Figure 7 shows the evolution of fitness. In the course of this test, none of the generations generated a robot capable of winning all battles against this rival.

In spite of this, one of the best-adapted robots was tested. This robot never fires; instead, it places itself in the edges of the arena, circulating at a speed that prevents chimpokomon from successfully hitting it, making its adversary eventually run out of energy (video available at https://youtu.be/dtZ_-bL2YqM). Though it won half its confrontations with the rival with which it was trained (*chimpokomon*), it won few battles against other rivals (see Table 5).

### 5.3. Third Instruction Set

This instruction set was based on the first. We made it more modular by creating instructions like “ifsee” to replace “ifmine”, “ifshoot”, etc. The “ifsee” instructions have a new terminal that can be “mine,” “robot”, “shot”, “cookie”, “wall”. If, for example, a user has a robot that shoots if it sees a mine, a child of this robot easily can “inherit” a simple mutation of this behavior and shoot if it sees a robot—see the examples of mutation (Figure 8) and crossover (Figure 9).

The nodes that represent the strength level of each action were also modified by defining three strength levels: “1” maximal strength, “1/2” medium strength, and “0” minimal strength, diverging from the first instruction set’s four and the second instruction set’s two strength levels.

Figure 10 shows the behavior tree of an example robot that is programmed under the third instruction set, as in Figure 4 and Figure 6, to destroy the mines. It rotates the radar at maximum speed, but fires with only half power. The node indicates that the radar rotates clockwise.

The selection evolutionary computation method was changed from “Greedy Over-Selection” to the simpler “Fitness Proportionate”. The parameters of this experiment are shown in Table 6.

The training battles were also changed. The experiment encompassed three stages, in which the evolved robot controllers face increasingly difficult opponents. The idea was to promote a suitable fitness landscape, by including tasks with different complexity levels and gradually increasing the “weight” of the most complex tasks. Additionally, by increasing the number of battles, the randomness elements of the score was decreased.

In the first stage, a total of eight training combats are performed, two each against the robots rogynt, pikachu, chimpokomon and falky. This stage proceeded until a robot controller able to win all 8 battles was found. The first row of Table 7 depicts the results obtained when such a robot conducted 50 battles against those opponents. It appeared the perfect score of eight wins was partially due to luck, and in fact the results of a larger set of battles against chimpokomon and falky revealed that the robot was not particularly successful against those adversaries.

In the second stage, 10 training combats per fitness evaluation were performed: four against chimpokomon (arguably the most difficult opponent), and two against each of the remaining robots (rogynt, pikachu and falky). This stage proceeded for seven generations. The results of the best robot in 50 battles against each adversary are shown in the second row of Figure 7; there was a remarkable increase in performance against chimpokomon.

Finally, in the third stage, which encompassed the final 10 generations, the number of training combats per evaluation was increased to 14 (six against chimpokomon, four against falky, two each against rogynt and pikachu). The results of the best robot in 50 battles against each adversary is described in the third row of Table 7; there was a significant increase in performance in battles against falky.

Figure 11 shows the evolution of fitness throughout the three stages of the experiment. Although the difficulty of the tasks increased from stage to stage, average fitness increased steadily throughout the course of the experiment (except a slight drop in the transition from stage 1 to 2). This indicates that the approach of gradually increasing the complexity of the task led to the emergence of robots that perform consistently well against a varied set of adversaries. Additionally, this instruction set provided us with a robot that won most of its battles against each rival. This approach, therefore, could be considered a solution in principle.

The behavior of the robots can be best understood by watching video of them in action (available at https://youtu.be/HGs7ka0MESI). The robot did not move at all—its performance results from its ability to track and aim. The robot fired continuously and used its shots for offensive and defensive purposes. In RTB, shots can collide and, as such, the evolved robot successfully eliminated several shots of its adversaries.

Although the robot was effective in the early location and the elimination of a mobile robot, it does possess some limitations. First, it does not move, and though this may not be necessary, it can be considered a desirable behavior. Second, and most importantly, when the robot sees a mine or cookie, it stops spinning and shooting, which leaves it at the mercy of its rival. As such, when the frequency of the appearance of mines and cookies is increased, this robot becomes an easy target (Table 8).

### 5.4. Fourth Instruction Set

The previous set was improved by adding a terminal and a function that created a short-term memory. The terminal was applied to the “if” functions and can have two values: “NOW” or “BEFORE”. Thus, comparisons are made to the current or previous state.

The new function “IFRELATIONAL” allows the robot to compare any data provided by RTB to the robot in two different cycles (the current and previous ones). Each robot receives a set of data during each cycle that includes radar information (distance to the object, type of object, radar angle), position of the robot (x, y, angle, velocity), level of energy, number of robots left, information about collision (object and angle), etc. The first terminal of “IFRELATIONAL” is a number between 1 and 27 that determines the data to compare. This function has also three child nodes, which will be activated if the parameter is less than, equal to, or greater than, relative to a numeric constant.

For example, the robot with this new instruction set knows if another robot entered the radar (i.e., was not picked up by radar before, but now is). Robots using the previous instruction sets only knew the current state. These robots can access other parameters managed by RTB that were not included in the previous instruction sets (e.g., their own energy levels). Table 9 outlines the function and terminal sets of this instruction set.

The behavior tree of a robot that is programmed to destroy the mines under the fourth instruction set (as in Figure 4, Figure 6 and Figure 10) is shown in Figure 12. This robot rotates the radar with maximum speed, but fires with only half of its power. The “NOW” node represents what takes place in the current state, not the previous one.

In previous experiments with this instruction set, and after a large number of generations, we obtained robots that were not able to win the majority of fights despite otherwise promising results. An analysis of results led us to believe that the problem was associated with the size of the search space, which was significantly increased by the inclusion of the function “IFRELATIONAL”, with 3 branches. To overcome this problem, we limited the maximum size of the tree to 11, only slightly above the maximum initial size of 9.

Similar to the previous experiment, initial fitness was assigned by fighting relatively simple adversaries. Once adequate results were found, tougher opponents were included in the fights. This procedure resulted in two stages. In the first stage, lasting 21 generations, fitness was calculated through 15 battles: seven against chimpokomon, three against falky and guzman and two against rotateandfire (an extremely simple battle robot distributed with RTB as an example). The second stage lasted 37 generations and fitness was calculated through 16 combats, four each against chimpokomon, falky, guzman and spock. Spock essentially replaced rotateandfire, which was no longer useful in this stage of evolution, and all adversaries fought the same number of battles. Table 10 shows all the parameters used in this experiment and the associated values.

Figure 13 depicts the evolution of fitness through the two stages of the experiment. The most relevant information is the average fitness, since there may be individuals who reach the maximum fitness in an experiment without constituting a valid solution, and there will always be an individual with the minimum fitness in each generation.

During the first generations, average fitness rapidly increased, mostly due to the presence of easy adversaries. From generation 21 onwards, a sharp jump in average fitness is shown, which is due to the fact that the fitness calculation changes. Fitness is calculated with much more complex robots. In the second stage, fitness evolved slowly but steadily. In final generation, the average fitness is above 60%, that is, more battles are won than are lost in the population as a whole (and with a very difficult fitness). This instruction set provided a valid robot, with more than acceptable results due to a large number of victories (Table 11).

To test the statistical consistency of the valid robot, new experiments were performed: 30 independent runs of 50 fights against each opponent (Table 11). It was performed only in this case due to the computational requirements (26 h of computation in total using an AMD Phenom II X4 955 Processor 3.2 Ghz GSkill 4 GB PC3-10666 DDR3-1333). Performing these tests during evolution would be impracticable. Table 12 illustrates the mean and statistical deviation from this experiment, in which the clear advantage is observed with respect to opponents, winning more than 74% of fights, with a standard deviation below 7% in the worst case.

Figure 14 depicts the code of the robot presented in Table 11. A video shows this robot fighting against KambfBot (available at https://youtu.be/kXEZ5oY_Lqs). In the image sequence of Figure 15, the behavior of the evolved robot can be observed in one of its battles against falky. The evolved robot’s strategy consists of moving toward the edge of the arena and then moving along the perimeter. When it sees a rival it fires. If the rival is sufficiently close, the robot moves away from the edge and closely turns around the enemy, distancing itself from the adversary while firing at it.

This behavior can be considered an evolution of the behaviors resulting from experiments 2 and 3. In the second experiment, the most effective evolved robots ran away from chimpokomon, avoiding its shots, but not firing in return. In the third experiment, the robot successfully tracked and aimed, but stood still. In the current experiment, evolution generated a robot that was successful in avoiding shots, tracking and aiming giving rise to complex and creative behavior (totally artificially generated). It is important to highlight that since RTB is a low level simulator and using very basic instructions and sensors, this same behavior could be adapted very simply to physical robots.

## 6. Facing Other Adversaries

Two final experiments were conducted: (i) checking whether the obtained robot achieved good generalization and the ability to defeat other robots not previously faced and (ii) a free-for-all facing all previously seen robots.

For the first experiment, the robot used was *jBot*, winner of the previous RTB Championship. It faces its opponent, moving toward it and firing multiple shots. If finds a cookie, tries to eat it, and it tries to destroy mines. This robot is free provided within the latest version of the simulator. The battles were performed following the same specifications seen previously, with a total of 50 independent battles. In this case, our robot won 36 of 50 attempts.

As discussed on Section 2, several studies have tried to use GP to generate single robots capable of defeating numerous adversaries at the same time without success. We used the same conditions as in previous experiments to test our solution robot against all previously seen robots in a free-for-all. Table 13 shows the results.

The results show a big difference between our solution robot and the rest; our evolved robot had the greatest number of wins and second place finishes. In addition, our robot placed in the last two positions the fewest times. These data indicate that the resulting robot is not only good at one-on-one fighting, but also in multi-tank struggles.

## 7. Conclusions and Future Developments

To solve a competition problem between robots, we used a complex system (GP) with low-level instructions (rotation and acceleration) rather than high-level instructions (such as attack or escape). This approach presents several differences from other state-of-the-art work in this area. 1. In general, Robocode is used in related studies. This system is currently more widespread in regular competitions. However, Robocode presents only a higher-level interface. For example, it provides the speed and direction of detected tanks, while RTB does not provide this information. Robocode also limits the maximum size of the robot’s compiled program to 1500 bytes, which restricts the behavioral capabilities of AI-generated robots and competing human-made robots. Finally, Robocode limits the languages that can be used for robot development. 2. Many state-of-the-art solutions use GP to solve some aspects of the creation of the robot. In [29,30] the robot strategy is defined by the authors, and GP is used only to define three numerical values. In the case of [24] it uses ANN to determine the angle to which the gun is turned and GP is used for movement and radar control. Hong and Cho [27] adopted genetic algorithms to produce different behavior styles based on six primitive behaviors.

3. The most similar examples are [22] and [34,35]. Eisenstein [22] was able to evolve Robocode players, each able to defeat a single opponent, but was not able to beat testing adversaries. In [34,35], the authors state that the behavior is “relatively simple”, consisting of wall following and firing “roughly”. 4. None of these systems were able to defeat all adversaries faced—though it is necessary to emphasize that the adversaries they faced differed from those presented in this article.

In this work, three different experiments that resulted in better versions of the robot are described. The first robot presented behavior similar to that of [35], while the others presented more complex behaviors. The final robot, having detected the other tank, revolves around it as it moves away and constantly shoots at it. The only information provided by the RTB is the distance and energy of the enemy robot, so the calculation of its future trajectory was carried out by the solution robot. This robot was able to defeat all its rivals, including in a multi-tank battle mode, representing a possible solution to a variable problem in which each battle is different: mines and the cookies appear in different places and at different times, the robots are arbitrarily placed in the arena at the start of the battle, the rival’s sequence of movement can vary from one battle to another, etc. We did not resort to any sort of pre-defined strategy or high-level function, nor did we supply additional information to the robots beyond that made available by the RTB environment during fights.

The behavior of evolved robots displays wide variety. In this paper, we report the outcomes of four experiments. However, we conducted a total of 47 experiments with the four instruction sets we describe. The complexity of the evolved behaviors significantly increases when short-term memory is included. For instance, the ability of the robot to move while tracking and aiming at its adversary was only achieved using short-term memory. Unfortunately, the fourth instruction set tends to generate programs (i.e., trees) that are very long, compromising evolvability. To overcome this difficulty we had to explicitly limit the size of the trees.

A major difficulty of this sort of experimentation is the computational cost involved. Although combat was performed in parallel (15 bouts at a time), these experiments still required 1043 CPU hours. To manage these computational costs we made two important decisions. First, we limited the number of runs of each experiment to one. Although this prevents us from making a detailed statistical analysis of the results, we consider that the experiments still demonstrate that interesting and effective behaviors can be found by evolutionary means. We are considering to apply new techniques for reduce the computational cost, such us microfluidic based devices [37,38]. Second, due to these costs, we did not explore the possibility of adopting co-evolutionary approaches, with evolved robots battling each other. Such experiments could further improve the results and increase their generalization abilities, since the robots would be fighting evolving adversaries and, therefore, would not be able to find shortcuts that yield good results against a specific bot but that are not generalizable. We plan to explore this line of research in the future.

Other techniques that could in principle be used to solve this problem are neural networks, although their use implies certain important difficulties, such as adapting the output of neurons to different robot operations (each operation has a set of different parameters).

## Figures and Tables

**Figure 1 entropy-22-01362-f001:**
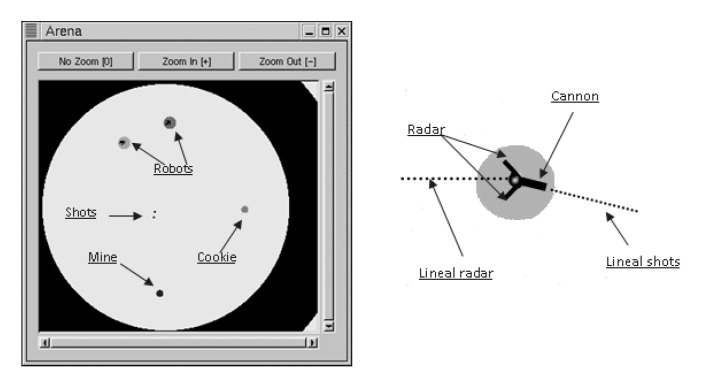
Representation of the battle arena and the robot’s components.

**Figure 2 entropy-22-01362-f002:**
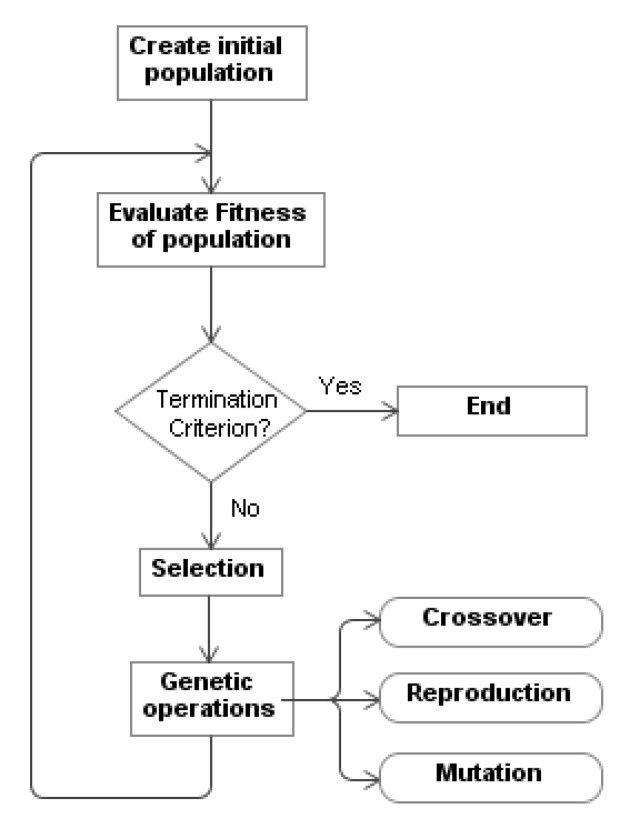
Flowchart of the genetic programming.

**Figure 3 entropy-22-01362-f003:**
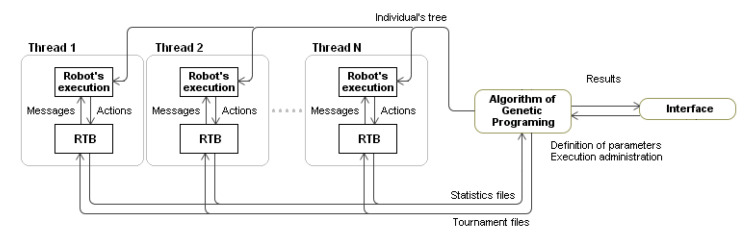
Diagram of the system and its interaction with the RealTimeBattle (RTB) platform. The Algorithm of Genetic Programming module control several executions of RTB each with a different individual of the population. The interface only allows the user to control the general parameters of the system.

**Figure 4 entropy-22-01362-f004:**
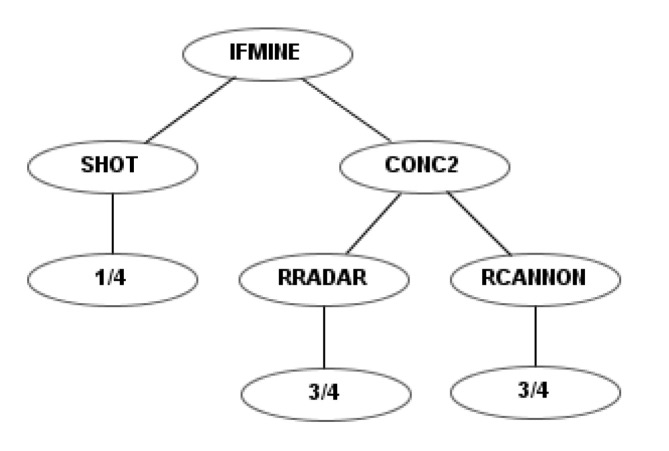
Example robot using the first instruction set.

**Figure 5 entropy-22-01362-f005:**
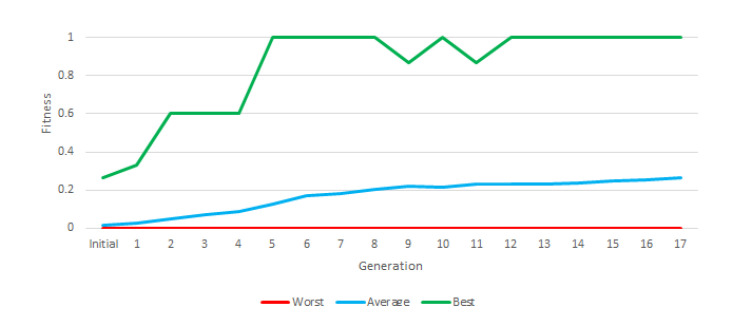
Best, worst and average fitness of each generation of the first instruction set.

**Figure 6 entropy-22-01362-f006:**
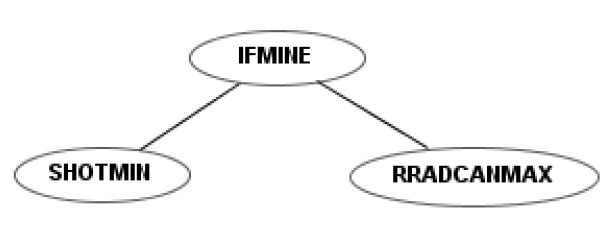
Example robot using the second instruction set.

**Figure 7 entropy-22-01362-f007:**
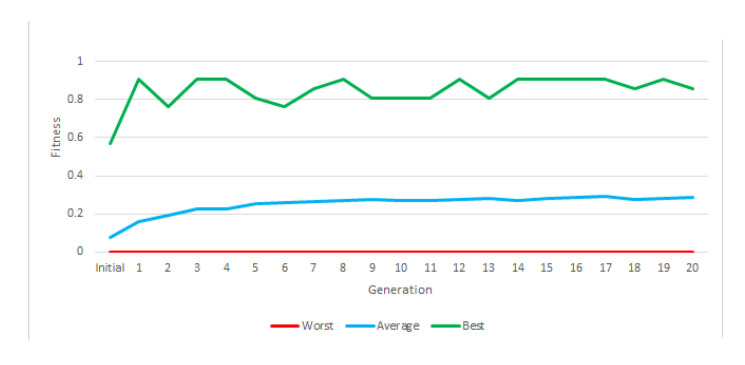
Graphic of the best, worst and average fitness of each generation of the second instruction set.

**Figure 8 entropy-22-01362-f008:**
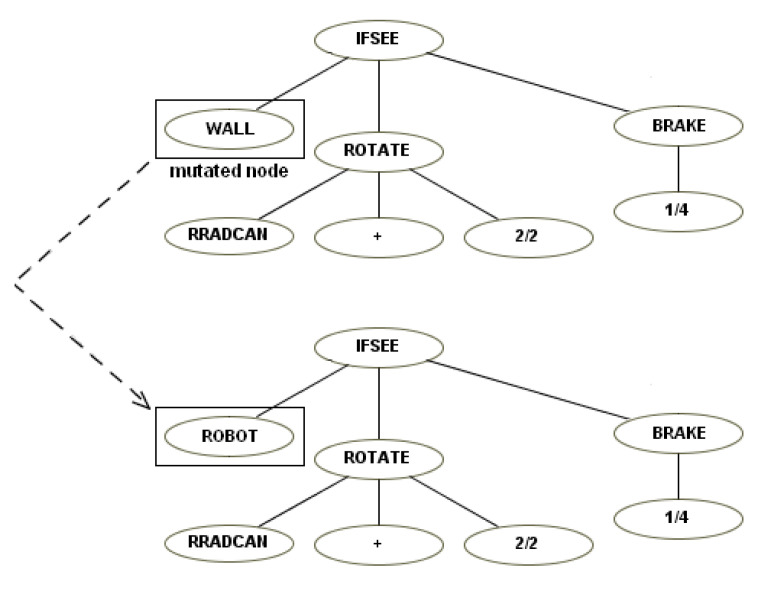
Example of mutation.

**Figure 9 entropy-22-01362-f009:**
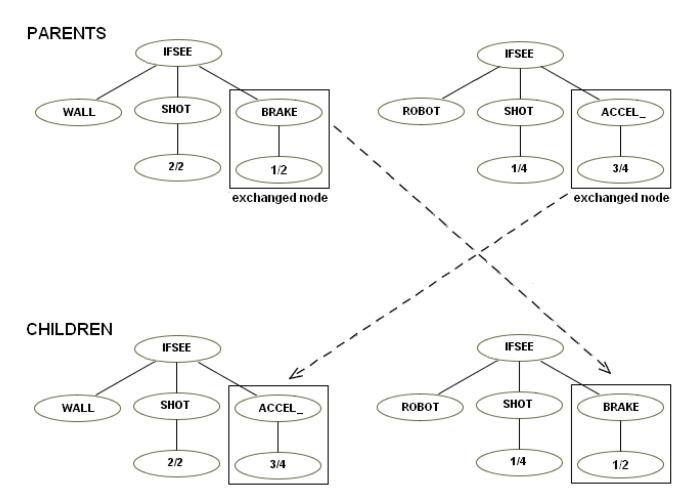
Example of crossover.

**Figure 10 entropy-22-01362-f010:**
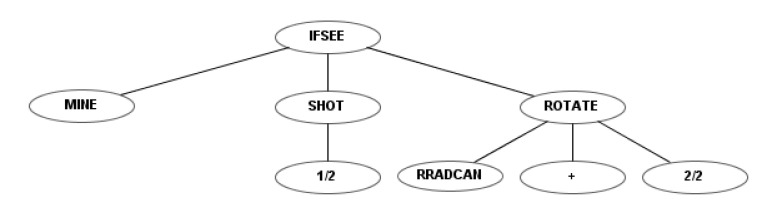
Example robot using the third instruction set.

**Figure 11 entropy-22-01362-f011:**
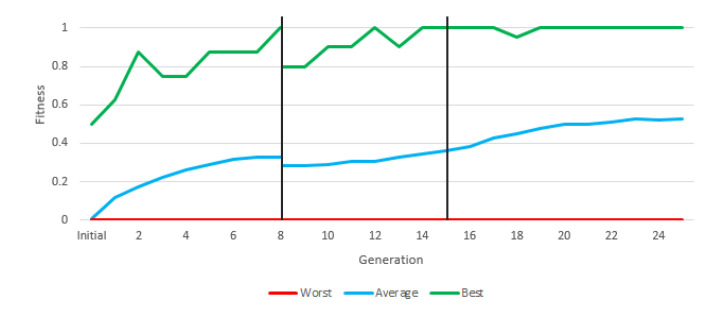
Graphic of the best, worst and average fitness of each generation of the third instruction set.

**Figure 12 entropy-22-01362-f012:**
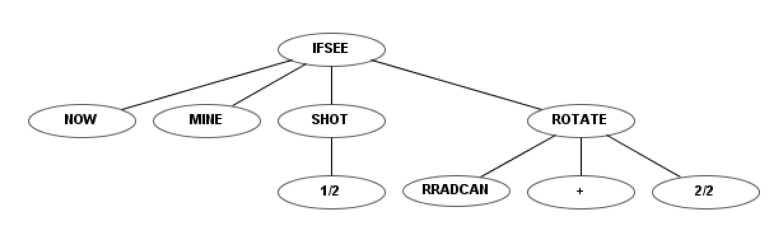
Example robot using the fourth instruction set.

**Figure 13 entropy-22-01362-f013:**
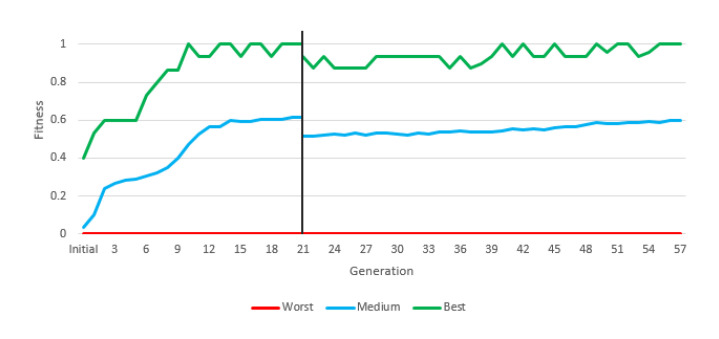
Training results of the fourth instruction set.

**Figure 14 entropy-22-01362-f014:**
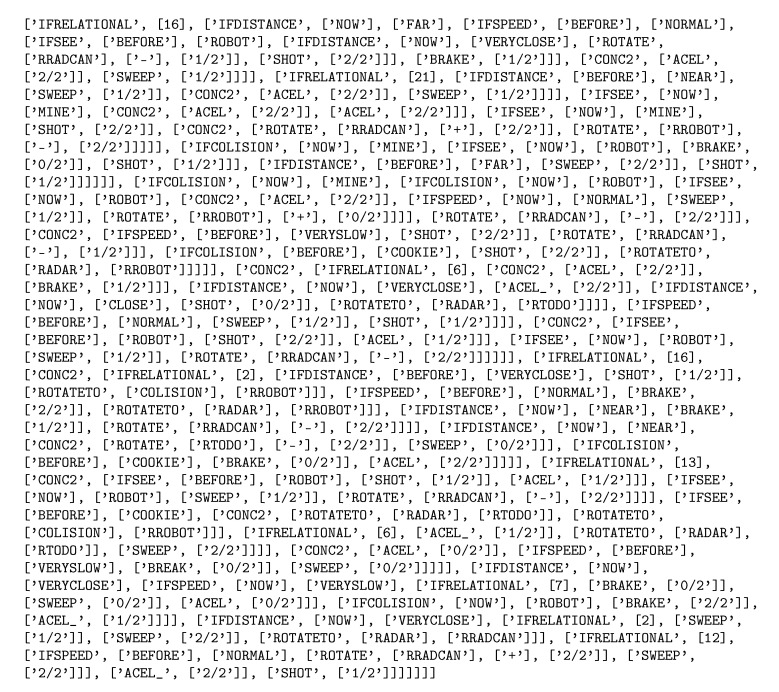
Code of the winning robot.

**Figure 15 entropy-22-01362-f015:**
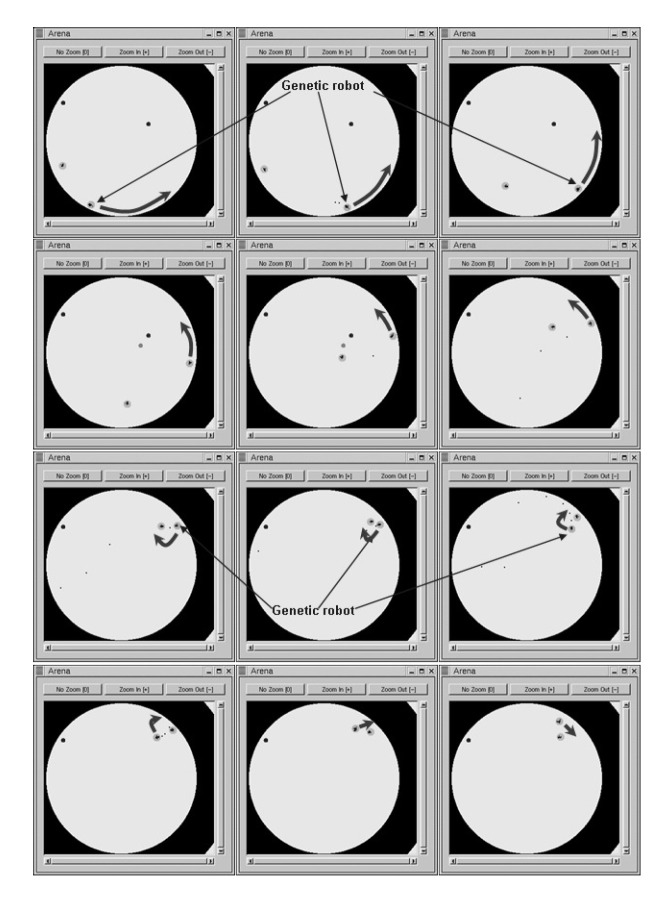
Images of a battle between our solution robot and falky (arrows indicate the movement of the solution).

**Table 1 entropy-22-01362-t001:** Functions defined in the first instruction set.

Functions	Description
Ifrobot, ifshot, ifwall,	Check if the observed object is a rival robot,
ifcookie, ifmine	a shot, a wall, a cookie or a mine.
Ifradle	Check if the distance at which the object is found is smaller than a determined distance.
Ifcollrobot, ifcollshot,	Check if the robot collided with a rival, was hit,
ifcollwall, ifcollcookie,	ran into the wall, ate a cookie or stepped on
ifcollmine	a mine.
Ifenle	Check if the energy of the robot is below a determined level.
Conc2	Execute the two son nodes of the tree.
Rradar	Rotate the radar clockwise.
Rradarm	Rotate the radar counterclockwise.
Rcannon	Rotate the cannon clockwise.
Rcannonm	Rotate the cannon counterclockwise.
Rrobot	Rotate the robot clockwise.
Rrobotm	Rotate the robot counterclockwise.
Rradarto	Turn the radar at a determined angle.
Rcannonto	Turn the cannon at a determined angle.
Swcannon	Set the cannon in sweeping mode.
Swradar	Set the radar in sweeping mode.
Rstopall	Hold all the rotations.
Shot	Fire a shot.
Accel	Accelerate.
Accel_	Retreat.
Brake	Brake.

**Table 2 entropy-22-01362-t002:** Parameters of the first instruction set.

Parameter	Value
Size of the robot populations	2000
Number of evaluated generations	17
Initial maximum height of the individuals	7
Allowed maximum height of the individuals	20
Generation method of the initial population	Ramped half-and-half
Reproduction probability	10%
Crossover probability	90%
Mutation probability	0%
Selection method of individuals	Greedy Over-Selection
Points for each battle won	3
Points for each battle drawn	1
Points for each battle lost	0
Battles	5 battles against each rival
Rivals	pikachu, chimpokomon, falky,
	rogynt

**Table 3 entropy-22-01362-t003:** Results of confronting the best four robots of generation 15 with the rivals that participated in the training 30 times.

RIVAL												
ROBOT	Pikachu			Chimpokomon			Falky			Rogynt		
	W	T	L	W	T	L	W	T	L	W	T	L
Robot 1	**3**	0	27	**0**	3	27	**2**	0	28	**8**	4	18
Robot 2	**2**	0	28	**1**	6	23	**0**	2	28	**10**	2	18
Robot 3	**1**	0	29	**0**	6	24	**1**	2	27	**7**	6	17
Robot 4	**0**	1	29	**1**	5	24	**0**	0	30	**8**	2	20

**Table 4 entropy-22-01362-t004:** Modified parameters of the second instruction set.

Parameter	Value
Size of the robot populations	1000
Number of evaluated generations	20
Battles	7 battles against *chimpokomon*

**Table 5 entropy-22-01362-t005:** Results of confronting 30 times the best robot obtained in generation 8 with the rivals that participated in the training in the previous experiment.

RIVAL												
ROBOT	Pikachu			Chimpokomon			Falky			Rogynt		
	W	T	L	W	T	L	W	T	L	W	T	L
1-generation 8	**0**	2	28	**15**	1	14	**3**	1	26	**0**	0	30

**Table 6 entropy-22-01362-t006:** Modified parameters of the third instruction set.

Parameter	Value
Size of the robot populations	2000
Number of evaluated generations	10
Initial maximum height of the individuals	6
Selection method of individuals	Fitness-Proportionate
Battles	14, six against *chimpokomon*,
	four against *falky* and two
	against *rogynt* and *pikachu*

**Table 7 entropy-22-01362-t007:** Results of the test with the best robot obtained in generation 9.

RIVAL												
ROBOT	Pikachu			Chimpokomon			Falky			Rogynt		
	W	T	L	W	T	L	W	T	L	W	T	L
1-generation 8	**30**	0	20	**10**	1	39	**19**	1	30	**25**	0	25
1-generation 15	**45**	0	5	**37**	1	12	**31**	0	19	**35**	0	15
1-generation 25	**45**	0	5	**36**	3	11	**46**	1	3	**37**	0	13

**Table 8 entropy-22-01362-t008:** Results of the test for the robot of Table 7, increasing the appearance of mines and cookies.

RIVAL												
ROBOT	Pikachu			Chimpokomon			Falky			Rogynt		
	W	T	L	W	T	L	W	T	L	W	T	L
1-generation 9	**33**	0	17	**19**	5	26	**14**	0	36	**20**	1	29

**Table 9 entropy-22-01362-t009:** Function and terminal sets defined in the fourth instruction set.

Functions	Description
IFCOLISION	Check if the robot has collided with an OBJECT.
	*(MOMENTUM, OBJECT, FUNCTION+TERMINAL, FUNCTION+TERMINAL)*
IFDISTANCE	Check if the distance at which an OBJECT is
	found is smaller than a determined distance.
	*(MOMENTUM, DISTANCE, FUNCTION+TERMINAL, FUNCTION+TERMINAL)*
IFRELATIONAL	Compare two data provided by the robot in two
	different MOMENTUM.
	*(POSITION, FUNCTION+TERMINAL, FUNCTION+TERMINAL,*
	*(FUNCTION+TERMINAL)*
IFSPEED	Check if the speed of the robot is slower than
	a specified MOMENTUM.
	*(QUANTITY)*
IFSEE	Check the observed OBJECT.
	*(MOMENTUM, OBJECT, FUNCTION+TERMINAL, FUNCTION+TERMINAL)*
CONC2	Execute the two child nodes of the tree.
	*(FUNCTION+TERMINAL, FUNCTION+TERMINAL)*
SWEEP	set the robot in sweeping mode.
	*(QUANTITY)*
ROTATETO	Rotate the robot at a determined angle.
	*(OBJECT, COMPONENT)*
ROTATE	Rotate a COMPONENT of the robot.
	*(COMPONENT, DIRECTION, QUANTITY)*
ACCEL	Accelerate.
	*(QUANTITY)*
BRAKE	Brake.
	*(QUANTITY)*
ACCEL_	Retreat.
	*(QUANTITY)*
Terminals	Possible values
DIRECTION	−,+
ROTATION	COLISION, RADAR
COMPONENT	RRADCAN, RROBOT, RALL
QUANTITY	0/2, 1/2, 2/2
MOMENTUM	NOW, BEFORE
POSITION	VERYCLOSE, CLOSE, NEAR, FAR
SPEED	VERYSLOW, SLOW, NORMAL, FAST
OBJECT	SHOT, COOKIE, MINE, RADAR, ROBOT

**Table 10 entropy-22-01362-t010:** Parameters of the fourth instruction set.

Parameter	Value
Size of the robot populations	1000
Initial maximum height of individuals	9
Allowed maximum height of individuals	11
Generation method of the initial population	Ramped half-and-half
Reproduction probability	10%
Crossover probability	90%
Mutation probability	0%
Selection method of individuals	Fitness-Proportionate

**Table 11 entropy-22-01362-t011:** Results of the test with the best robot obtained in generation 37.

RIVAL												
ROBOT	Pikachu			Chimpokomon			Falky			Rogynt		
	W	T	L	W	T	L	W	T	L	W	T	L
1-generation 37	**47**	0	3	**42**	0	8	**33**	1	16	**35**	0	15

**Table 12 entropy-22-01362-t012:** Results of 39 independent runs, each run comprising 50 combats with each rival robot

Robot	Average of Wins	STD
Pikachu	93.60%	3.30%
Chimpokomon	78.00%	5.75%
Falky	74.00%	6.56%
Rogyint	74.53%	5.87%

**Table 13 entropy-22-01362-t013:** Results of 50 battles. Each battle involves five robots. The first one is the best robot obtained in our experiments; the last is the winner of the previous RTB Championship. The first row shows the number of times each robot obtained first place out of the 50 battles. The second row shows the number of times each robot obtained second place, and so on.

Position	Our Best Robot	Pikacku	Chimpokomon	Falky	Rogynt	jBot
1st Place	14	10	10	6	3	7
2nd Place	12	9	5	8	7	9
3rd Place	8	7	3	11	9	12
4th Place	10	10	8	13	5	4
5th Place	2	8	9	5	13	13
6th Place	4	6	15	7	13	5

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
