# Peer review of "A Genetic Programming-Based Low-Level Instructions Robot for Realtimebattle"

_entropy, 2020, doi:10.3390/e22121362_

Round 1
Reviewer 1 Report
THe paper has been improved and it is of good quality.
Reviewer 2 Report
The authors should revise this manuscript according to the following points:
1.In section of Introduction, the authors should supplement some recent relevant literature.
2.In section 2, the authors should elaborate on Figure 3.
3.In section 4, the authors should highlight the peculiarities of their work in the Evolution toward a competitive robot.
Reviewer 3 Report
No
This manuscript is a resubmission of an earlier submission. The following is a list of the peer review reports and author responses from that submission.
Round 1
Reviewer 1 Report
The paper deals with designing controls for a robot involved in robotic tank fights. The controllers are designed using a genetic programming (GP) approach. The robotic tank fight used is RealTimeBattle (RBT). The control is evolved using GP, which takes as a fitness evaluation the average performance of each evolved candidate against five human-coded rivals. The paper describes the RTB domain, discusses the GP implementation, and presents computational experiments and results. In general, this might be an interesting paper using a well-known evolutionary computation algorithm for solving a well-defined optimization problem. However, I think there are some serious shortcomings that a revised version of the paper needs to address before it can be reconsidered for possible publication in Entropy.
I am highly puzzled about the choice of RTB as fight environment as RTB is kind of out-dated. The last competitions, from which the human-coded rivals have been taken, was in 2002, which is almost 20 years ago. Moreover, with Robocode there is an alternative, for which results from alternative GP implementations have been reported. Against these implementation the approach of the paper could be tested, see for instance: GP-robocode: Using genetic programming to evolve robocode players. Shichel et al. Conference on Genetic Programming, 2005; Evolved to Win, Sipper, 2011; Co-evolving robocode tanks. Harper, Proc GECCO 2011; Evolving robocode tanks for Evo robocode. Harper, Genetic Programming and Evolvable Machines, 2014.
My feeling is that the RTB environment and the selected human-coded rivals do not give a particularly competitive setting. The best fitness for all instructions sets is reached after a rather small number of generations. I would like to see the behavior for a longer run time, and a lower population size. The current setting looks to me as rather promoting a random search than an evolutionary process.
The references are not up-to-date and do not represent the state-of-the-art in GP based robotic fight application. The ``newest’’ works are from 2010 and 2011, and McCarthy’s 1960 paper about recursive functions was surely a milestone in the (pre-)history of machine learning, but to cite it in a 2020 paper about GP and robotic fight applications is just strange, I feel. What is more important, from the references, a reader could get the impression that using GP for designing controls of tank fighting robots is an area where not forerunner works exists, which is far from the truth, see also the main point of my review.
Some minor points:
- The abstracts announces GP as a complex system technique. I think this is awkward. Surely, such a categorization is probably not completely outlandish, but a more common view is that GP is a part of evolutionary computation, which is a part of computational intelligence.
- The introduction sets out with praising David Fogel’s book ``Blondie24’’ as a hallmark for learning chess without chess knowledge being programmed into the game engine. This is not completely undeserved, but the book is from 2002 and in the light of the progress made by AlphaZero and clones in the last ten years, it is, I think, largely out of proportion. I recommend reconsidering the ``opening’’.
- There are a huge amount of single-sentence paragraphs, which makes reading the paper rather difficult. I recommend to rewriting the paper.
Author Response
Thank you very much for your suggestions. We think that it allow us to really improve the paper.
About the state of the art in EC applied to RTB and robocode, we were surprised because two of the references proposed by you was already included and commented
on the paper: GP-robocode: Using genetic programming to evolve robocode players". (ref 22) and "Co-evolving robocode tanks" (ref 23). The third one, the book "Evolve to win"
have a chapter on GP applied to Robocode that is based on paper "GP-robocode: Using genetic programming to evolve robocode players" (ref 22), so we include it on the paper. We also
included the paper "Evolving robocode tanks for Evo robocode" that is an exteded version of the paper "Co-evolving Robocode tanks.".
We uptated some information about all papers, that facilitate the comparison with our proposal.
Moreover, in order to made more clear the differences betweeen SOA and our proposal, we included this paragraph on the conclusions:
" This paper presents several differences with the other works of the state of the art.
1.- In general, Robocode is used in the other jobs. This system is currently more widespread with regular competitions. On the other hand, Robocode presents a less low-level interface. For example, it provides the speed and direction of detected tanks, while RTB does not provide this information. It also limits the maximum size of the robot's compiled program to 1500 bytes, which restricts the behavioral capabilities of AI-generated robots and competing human-made robots. Finally, Robocode limits the languages ​​that can be used for robot development.
2.- Many of the works of the state of the art use GP to solve partial aspects of the creation of the robot. in [book and 2005] the robot strategy is defined by the authors and GP is used only to define three numerical values. In the case of [GADE] it uses ANN to determine the angle to which the gun is turned and GP to the movement and radar control. Hong and Cho [20] adopted genetic algorithms to produce different behavior styles based on six primitive behaviors.
3.- The most similar examples are [eisten] and [harper]. Eisenstein was able to evolve Robocode players, each able to defeat a single opponent, but was not able to
beat testing adversaries. In [HARPER2014] the authors state that the behavior is "relatively simple", consisting of wall following and firing "roughtly".
4.- None of these systems was able to defeat all its adversaries. Although it is necessary to emphasize that the adversaries they faced are different from those presented in this article.
In this work three different experiments are shown, in which better versions of the robot are obtained. The first robot obtained on this paper presents a behavior similar to that of [HARPER2014] while the others present more complex behaviors. The final robot, having detected the other tank, surrounds it as it moves away and constantly shoots at it. In this context, it is necessary to emphasize that the only information provided by the RTB is the distance and energy of the robot, so the calculation of its future trajectory is carried out by the robot. "
About the use of RTB, we understand your point about being old. However, RTB mantain some serious advantages that we showed now in the paper:
- It allow to use any language (including python), so facilitating the design.
- More relevant: it does not limit the program size (as Robocode do), so not artificial limit to the possibilities of GP behaviour. Robocode limits the compiled code to 250, 750 or 1500 bytes. Of course, this also limits the capabilities of human programmed robots.
- Finally, there are some differences that make Robocode higher level than RTB. The most important is that in Robocode, tanks can detect other tanks, their energy level, speed and direction. In RTB, the tanks only detect the presence of other tanks, their energy and their distance (they do not know their speed or direction) which makes it much more difficult (and realistic) to track the tank.
We cannot agree about the lack of competitive setting in our paper.
In fact, we consider that the paper explores very well the relevance of the instructions set.
It shows 3 experiments that give raise to BAD results and one finally working.
In this paper, the fitness function is based on a limited number of combats (in some cases 20), and the characteristics of RTB and Robocode
made very random the combats difficulty. (paraphrasing [harper 2014]: "In a Robocode match the tanks are placed in a random position, with random
orientation. This can give a large advantage to one of the tanks. Only by running multiple matches can a fair view of a tank’s prowess against an opponent be
determined."). So, to obtain one individual (in a population of 2000) that wins all the combats is really easy.
As pointed on in other papers, the relevant value in these figures is the average fitness.
But, as we know the low number of combats in the fitness function, we tested the best robots with far more combats (e.g. 120 combats). That is shown on the paper on tables: 3, 5,7, 8 and 11.
As you can see in these tables, only the robots of the last set of rules win all four robots.
In the previous sets of rules, the best robots of the last generation losses more combats than wins.
About to see the results with even lower population time and longer run time, we will test it more deeply in the future. We had some experiments with lower population size, obtaning very bad results. But, we will insist on it. However, we cannot consider a population of 2000 individuals a high population size in GP. It is indeed a low population size. Even in the paper "Harper2014" suggested by you it can be seen :"Assuming a relatively low GP population of 1,000".
In the paper we show only several experiments but we did far more testes before we find the way we obtained the desires results. There are more than 6000 hours of machine time in all the experiments we did. This is mainly to the need to test each individual with several combats that should be done in real-time even if we made it with a high degree of paralelization.
About the references are not up-to-date, we agree, so we include the [Harper2014] and [Evolve to Win] references, as you suggest. We also delete the McCarthy’s 1960 reference.
What surprised us a little bit is your comment: "a reader could get the impression that using GP for designing controls of tank fighting robots is an area where not forerunner works exists, which is far from the truth, see also the main point of my review". The fact is that the original version of the paper includes 5 references of preious papers. You proposed 4 references: 2 already included in the paper and the other 2, updated versions of papers included in the paper. Anyway, the current version is better than the orignal one and make more clear the differences between the papers.
About minor points:
1.- Indeed. We announced GP as a complex system technique. We corrected this, so now is part of Evolutionary Computation (and in another place is announced as an AI technique).
2.- We included also a comment on the new advances in AI on chess and AlphaZero.
3.- We also agree. We did a profound English revision.
Finally, we would like to note on that the paper shows the different behaviors achieved by the robots through the experiments. We consider these samples of complex behavior, obtained through an AI technique without collaboration with humans and through low-level tasks, very relevant for a publication in Entropy journal.
best!
Juan
Reviewer 2 Report
This manuscript needs the following refinements:
1.In section of Introduction, the authors need to supplement some recent relevant literature.
2.In section 2, the authors should explain the advantages of genetic programming in solving this problem.
3.In section 4 and 5, the authors should explain in detail the results of the experiments and indicate what the methods other than genetic programming can be used.
Author Response
Dear reviewer,
Thank you so much for the comments. We have improved the article following your advice.
1.- We have not found new works on simulated tanks and evolutionary computing. We believe that due to some limitations of Robocode and not so good results in previous work, the area is not very dynamic. We hope that this article can reverse this situation.
However, we add some very recent articles on GP systems used in robot behavior and include some relevant references on evolutionary robotics.
2.- We explain the advantages of GP in solving this type of problem, among them:
- The structure and parameters of the solution are explored simultaneously.
- Has the ability to create comprehensive operator-based behaviors that can be tailored to a particular problem
-The output of a GP system is a program that can be analyzed
- GP is well suited to complex domains where little information is available
The GP system can even lead to behaviors beyond the programming capabilities of its creators and generate creative results, like some patents already created by GP systems.
3.- We explore other techniques that could be used for this task such as Artificial Neural Networks in conclusions. In addition, we made several changes to section 4 to explain the results obtained in more detail.
Reviewer 3 Report
The paper discusses about genetic programming for low level instructions for networks of robots. The evolutionary based algorithms for this aims are suitable. The authors develop an interesting procedure that allow to get to an easy instruction set. THe performance to achieve simplicity is welcome and in my opinion the author achieve a good task.
My curiosity is the implementation of real networks ao nano-microrobots really and the use of new technologies to realize it. It is outstanding to look at microfluidics based devices and approaches. Moreover in this are, also computational problems could arises or microfluidic devices could be used for making computations. Therefore I invite the authors to include their research in a more wide area. I therefore suggest to include in the reference the following contributions:
Microfluidics and NanofluidicsVolume 18, Issue 2, 2014, Pages 305-321Computational models in microfluidic bubble logic(Article)
- Anandan, P.,
- Gagliano, S.,
- Bucolo, M.Email Author
- View Correspondence (jump link)
EnergiesOpen AccessVolume 12, Issue 13, 2019, Article number 2556
A real time feed forward control of slug flow in microchannels(Article)(Open Access)
- Gagliano, S.,
- Cairone, F.,
- Amenta, A.,
- Bucolo, M.Email Author
- View Correspondence (jump link)
I think that the authors must be joint their studies with advanced topic in order to make thair contribution more appealing.
Author Response
Dear reviewer,
Thank you very much for your review.
We did not know about microfluidic based devices. Very interesting approach.
We included the citations you mentioned in the conclusions section.
best!
Juan
Round 2
Reviewer 1 Report
My main concern was not addressed in the revised version of the paper, which is to test the GP approach considered in the paper against existing GP approaches to robotic tank fights. Thus, I have the feeling that the results of the paper stand by themselves and have no connection to previous works in the field. Thus, they are hard to judge and give little insight into the working principles of EA techniques in a clearly defined domain of application. Besides, why the knowledge of speed and direction of detected tanks, which is provided by Robocode, should be a disadvantage, while not having these information, as in RTB, is a positive and unique feature, is not understandable to me.
My feeling is that the paper in its current version does not provide results as significant as required in Entropy, and I do not recommend publication.
Author Response
Dear reviewer,
We can see your point. Our idea is that the limitations of Robocode (especially the limitation in the size of the program up to 1500bytes) have led to much improvement in previous work. These results and the limitations of Robocode have made this line of research not too dynamic.
By using another simulator, in this case RTB, we can avoid these limitations and obtain more interesting results. It is important to note that:
- none of the state-of-the-art papers present complex behaviors
- none of the other papers are able to beat all the opponents presented
- the few who try multiplayer battles get very bad results.
Finally, having information on the speed and direction of the other robots is not a limitation of Robocode, obviously. But it is if you want a low level system. In a physical implementation of the robot, you could know the distance with respect to other adversaries but not the speed and direction. The absence of this information forces the programmed robot to define that speed and direction if it wants to use it.